# Enhancement of Confinement in Scaled RC Columns using Steel Fibers Extracted from Scrap Tyres

**DOI:** 10.3390/ma15093219

**Published:** 2022-04-29

**Authors:** Izaz Ahmad, Mudasir Iqbal, Asim Abbas, Yasir Irfan Badrashi, Arshad Jamal, Shahid Ullah, Ahmed M. Yosri, Moustafa Hamad

**Affiliations:** 1Department of Civil Engineering, University of Engineering & Technology, Peshawar 25120, Pakistan; izazahmad@uetpeshawar.edu.pk (I.A.); asimabbas@uetpeshawar.edu.pk (A.A.); yasir_badrashi@uetpeshawar.edu.pk (Y.I.B.); shahid.ullah@uetpeshawar.edu.pk (S.U.); 2Shanghai Key Laboratory for Digital Maintenance of Buildings and Infrastructure, School of Naval Architecture, Ocean and Civil Engineering, Shanghai Jiao Tong University, Shanghai 200240, China; 3Transportation and Traffic Engineering Department, College of Engineering, Imam Abdulrahman Bin Faisal University, P.O. Box 1982, Dammam 31451, Saudi Arabia; ajjamal@iau.edu.sa; 4Department of Civil Engineering, College of Engineering, Jouf University, Sakakah 72388, Saudi Arabia; 5Civil Engineering Department, Faculty of Engineering, Delta University for Science and Technology, Belkas 11152, Egypt; m.saad@bu.edu.sa; 6Department of Civil Engineering, College of Engineering, Al Baha University, Al Baha 65731, Saudi Arabia

**Keywords:** reinforced concrete columns, steel fibers, confinement, compressive strength

## Abstract

Steel fibers are widely extracted from scrap tyres, causing environmental concerns. This paper presents the use of steel fibers in variable proportions extracted from scrap tyres. The enhancement of the confinement was envisaged through the addition of steel fibers obtained from scrap tyres. The study included an experimental program for the development of constitutive material models for ordinary Portland cement (OPC) concrete and concrete with added steel fibers. A mix design was carried out for OPC, targeting a compressive strength of 3000 psi. Steel fibers were added to OPC in ratios of 1.0% to 3.0%, with an increment of 0.5%. Concrete columns, with cross-sectional dimensions of 6 × 6 inches and a length of 30 inches, were cast with both OPC and fiber-reinforced concrete. The column confinement was evaluated with a different spacing of ties (3- and 4-inch center-to-center). Compression tests on the concrete columns indicate that the addition of steel fibers to a concrete matrix results in an appreciable increase in strength and ductility. Overall, increasing the percentage of steel fibers increased the compression strength and the ductility of concrete. The maximum strain in the concrete containing 2.5% steel fibers increased by 285% as compared to the concrete containing 1% of steel fibers. An optimum percentage of 2.5% steel fibers added to the concrete resulted in a 39% increase in compressive strength, accompanied by a significant improvement in ductility. The optimum content of steel fibers, when used in confined columns, showed that confined compression strength increased with the addition of steel fibers. However, it is recommended that additional columns on the basis of the optimum steel fiber content shall be tested to evaluate their effectiveness in reducing the stirrup spacing.

## 1. Introduction

Concrete is a widely used construction material worldwide. A steady rise has been observed in the proportion of reinforced concrete buildings in the building stocks of both the developing and developed countries. The column is a critical member in a reinforced concrete structural system, playing a major role in the load transfer mechanism. The columns are primarily subjected to significant compression forces in megastructures, such as tall and long-span structures. However, compression is often accompanied by bending, either along one axis or both axes. Reinforced concrete columns may fail due to overloading during natural disasters, such as earthquakes, accidental lateral loads, and fire. The failure of a column in a reinforced concrete building system has the potential to result in a collapse mechanism; contrary to the failure of beams, which can only cause localized failure and do not jeopardize the structural integrity. The behavior of a reinforced concrete column is highly influenced by the confinement available to the column core. A well-confined column can significantly improve the load-carrying capacity and ductility. The confinement of concrete is actually the lateral restraint available to the concrete resulting from internal reinforcement, such as steel stirrups, spirals, or fibers, and by any external means, such as wrapping steel plates and carbon fiber-reinforced polymer (CFRP).

Two approaches are generally adopted for the provision of confinement in concrete columns, i.e., the traditional and the modern method. In the traditional method, lateral reinforcement in the form of hoops, cross-ties, or spirals plays an important role in the confinement of concrete in columns. It prevents the highly stressed longitudinal bars from buckling, increasing the axial capacity of the column and confines concrete, thereby increasing the ductility of a column and decreasing the spacing of transverse steel, improving confinement [1]. However, it also results in the discontinuity of core concrete and covers concrete, thus creating a plane of weakness. Additional overlapping hoops and ties can be used to overcome this weakness; however, it results in difficulty in the casting of the column due to excessive reinforcement, hindering the pouring of concrete, and, more importantly, the necessary vibration required to compact the concrete [2,3]. Saatcioglu et al. [4] and Kusuma et al. [5] carried out research to reduce reinforcement congestion by using welded reinforcement grids; however, the problem of discontinuity of core concrete and cover concrete was not resolved.

A variety of industrial wastes are used to supplement the properties of fresh and hardened concrete [6]. Rajamony Laila Gurupatham [7] investigated the influence of non-biodegradable granite Pulver on the mechanical and durability performance of self-compacting concrete. In the modern method of confinement, steel fibers, and polypropylene fibers (PPF) have been widely used for the confinement of concrete. A number of studies have been performed on different types of fiber-reinforced concrete (FRC). The studies concluded that FRC has good mechanical properties, such as compressive strength, tensile strength, and bending characteristics. It has been demonstrated that FRC increases the confinement of concrete in columns and also maintains the continuity between core and cover concrete. Furthermore, FRC helps in controlling the propagation of microcracks and prevents explosive spalling of concrete when exposed to fire [8,9,10,11,12]. Steel fibers contribute to the mechanical properties of concrete to a different degree depending upon its aspect ratio. Small fibers have a significant effect on compressive strength while slightly affecting tensile strength. Large fibers have the opposite effect. Similarly, longer fibers harden the reinforced concrete behavior as compared to short fibers, which makes it soften. Further fibers enhance the bending stiffness and affect the cracking pattern as compared to normal concrete [13,14].

Confinement through fiber-reinforced polymers (FRP) is done either externally or internally. In external confinement, steel plates and carbon fiber-reinforced polymers (CFRP) are wrapped around the RC column to enhance the strength capacity under axial compression. Externally bonded CFRP/GFRP hoop wraps provide confinement to concrete and lateral support to the longitudinal fibers and thus increase the strength of slender RC columns [15,16,17,18,19,20,21,22]. The repair, rehabilitation, and strengthening of existing structures have become a major part of the construction. However, even though this technique is successful in practice, it results in problems, such as additional weight, corrosion of steel plates, a need for skilled labor, and higher costs [8]. To avoid the issue of corrosion, CFRP fibers are widely used as wrapping material. In internal confinement, synthetic fibers such as steel fibers, polypropylene fiber, and glass fibers have been used in concrete to enhance RC column confinement and control microcrack propagation by the bridge mechanism. Currently, about 300,000,000 metric tons of fibers are used for the reinforcement of concrete. Steel fiber is the most used fiber (about 50% of the total tonnage used) followed by polypropylene (about 20%), glass (about 5%), and other fibers 25% [23]. The use of fibers in concrete can significantly improve its toughness, durability, tensile strength, and impact resistance. The addition of fiber in concrete also influences its mechanical properties, which depend upon the percentage and type of fiber. The main role of fiber is the bridging action, which controls crack propagation and leads to more post-cracking resistance [24,25,26,27]. Giuseppe Campione et al. [28] performed an experimental investigation to evaluate the confinement effect of steel fibers on the behavior of square RC columns. A total of sixteen columns were tested under axial compression and the test results revealed that steel fiber confinement enhances the structural performance of RC columns, especially concerning ductility. Raman Bharti et al. [27], in their work, tested a total of four columns joints, of which two were conventional specimens with different stirrup ratios and two with FRP wrapping. It was concluded that the fiber bridging mechanism is effective, as the steel fiber-reinforced concrete demonstrated better stiffness degradation over critically detailed specimens. Sidney Mindess et al. [28] investigated the influence of short carbon fibers in different percentages and of different lengths, both in the presence and in the absence of traditional steel spiral reinforcement. The results showed that there was no significant change in strength. However, a significant increase in energy absorption and maximum strain capability was observed.

The results of all the discussed research are significant, however, the use of these synthetic fibers has proved to be uneconomical. The use of synthetic fibers in concrete is well investigated by a variety of researchers; however, it is argued that synthetic fibers are expensive, with 1% of steel fiber addition approximately resulting in double the material cost of concrete. This has paved the way for exploring other sources of fibers that are economically feasible. Currently, in the European Union, about 600,000 tons of tyres are disposed to landfill each year (ETRA, 2006; FISE UNIRE 2007) [29]. Hence, steel fibers extracted from waste tyres provide an alternate fiber material as it is not only economical but also solves the environmental issue associated with the disposal of waste tyres. Therefore, the current research investigates the effect of using steel fibers on the compression behavior of concrete. It is important to mention that the current research implied the maximum percentage of steel as 3 percent in integration with concrete, thus initiating composite action of steel incorporated concrete. It is assumed that steel fiber in concrete would lead to corrosion; however, it is also known that the composite material increases the water tightness of concrete by resisting the elongation of microcracks to a macro level, thus preventing the moisture absorption to a greater extent. This way, corrosion resistance may be prevented, however, a detailed study is needed in this regard. If the corrosion is aggravated by incorporating steel fibers, the mix investigated in the current study can be used as inner concrete in construction.

## 2. Materials

### 2.1. Cement

Ordinary/commercial Portland cement was used as a binding material. A specific gravity test of the cement was conducted according to ASTM-C118 [30]. The fineness modulus of the cement was found to be in accordance with the requirements of ASTM C-184-94 [31]. The properties of the cement are listed in Table 1.

### 2.2. Coarse Aggregates

The locally available crushed stones were used as coarse aggregate in this study. The coarse aggregate was tested according to ASTM C-33 [32] and ASTM C-127 [33] for the determination of various properties listed in Table 2. The sieve analysis result is shown in a gradation curve in Figure 1. The gradation curve for the coarse aggregate used in this study was within the upper and lower limits, as defined by ASTM.

### 2.3. Fine Aggregate

Local natural sand was used as fine aggregate for the preparation of the experimental models. Fine aggregate was tested according to ASTM C-33 [32] and ASTM C-128 [34] for the determination of various properties listed in Table 2. The sieve analysis results are shown in the form of a grading curve in Figure 2.

### 2.4. Steel Fiber

The steel fibers extracted from used tyres, shown in Figure 3, were selected to be used as fiber reinforcement because of their superior mechanical properties. The steel fibers were subjected to tensile force in accordance with ASTM A 370 [35] tests for the determination of their tensile strength. The properties of steel fibers are listed in Table 3. It can be seen that 3 inches (75 mm) have been used in the preparation of concrete samples. The tensile capacity of the fibers is observed as 132 kg/mm^2^. The length of a fiber cut in 75 mm has been decided from the trial procedure (varying the size of fiber from 25–100 mm). These variable lengths of fibers were used in concrete and the corresponding slump was also monitored, yielding an acceptable slump at 75 mm.

### 2.5. Main Reinforcement (Rebars)

Grade 40 steel (having 40 ksi yield strength) was used as rebars. The #3 (10 mm) bar was used in longitudinal reinforcement, while the #2 (6.35 mm) bar was used in transverse reinforcement.

## 3. Mix Proportioning of Concrete

Concrete mix proportion was carried out according to ACI-211.1 [36]. The concrete was designed for a target compressive strength of 3000 psi based on the weight approach of mix proportioning. The detail of mix design/mix proportioning is listed in Table 4.

## 4. Specimen Preparation

### 4.1. Cylinder Preparation

A total of eighteen cylinders having a 6-inch (15 cm) diameter and 12-inch (30 cm) height were prepared for this study. These cylinders were divided into six groups on the basis of the percentage of steel fiber. Each group contained three specimens having the same percentage of steel fibers by weight. Group#1 (G1) represented the conventional concrete having no steel fibers, G2 contained samples having 1% steel fiber, G3 had 1.5% of steel fiber, G4 had 2% of steel fibers, G5 had 2.5% of steel fibers, and G6 had 3% of steel fibers as fiber reinforcement. Further, 3% was used as the maximum percentage of steel fibers as the slump value was approaching zero beyond 3%. For all the mix proportions, cement, aggregates, and steel fibers were weighed in a dry state and mixed in an electric mix machine. Water was added to the dry mixture and mixing was carried out for 200 s. After mixing, the molds were filled with concrete in three layers with each layer receiving 25 numbers of blows as per the ASTM standard specifications. After 24 h, the samples were de-molded and the cylinder specimens were wet-cured for 28 days. The studied concrete specimen is shown in Figure 4.

### 4.2. Columns Preparation (Confined Concrete)

A total of nine column specimens having a square cross-section of 6 × 6 inches (15 cm × 15 cm) and a length of 2.5 ft (76 cm) were prepared. All the specimens contained four longitudinal bars at their corners, with different spacing of the stirrups having a clear cover of 1 inch (2.5 cm). Column specimens were divided into three groups (C1, C2, and C3), each containing three specimens. C1 consisted of column specimens having 4 inches of center-to-center spacing of stirrups, the C2 group had a 3-inch stirrup spacing, while C3 contained column specimens having 4-inch stirrup spacing along with 2.5% of steel fiber. Confined concrete samples were cast only for 2.5% of steel fibers as this percentage gives the best results in the case of unconfined concrete tested in this study. The column samples were prepared using the same protocol as the one used in the preparation of concrete cylinders and described in the previous section. The column specimens are shown in Figure 5.

## 5. Experimental Setup for Fresh and Hardened Concrete

### 5.1. Slump Cone Test

Fresh concrete was tested for workability/consistency by the slump cone test in accordance with ASTM C143 [37]. The properties of fresh concrete are listed in Table 5, and test steps are shown in Figure 6.

### 5.2. Compressive Strength Test of Confined (Short RC Columns) and Unconfined Concrete Specimen

Concrete specimens were tested under axial compression in a universal testing machine (UTM), having an internal high precision pressure load cell (class 1) with a load measuring capacity equal to 2000 kN. Every specimen was loaded continuously until failure with a uniform loading rate. The displacement control technique was used to operate the UTM with a loading rate equal to ~0.2 mm/min. The specimens were placed in the jaws of the UTM, and care was taken to ensure that their centerline was exactly in line with the axis of the machine. The specimens were instrumented to measure the axial strain/deformation. The compressometer was used to measure strains in concrete cylinders according to ASTM C469 [38] while the strain gauge-based displacement transducers (Kyowa DT-50A, Kyowa, Singapore) measured vertical deformations under monotonic axial concentric loading. Steel plates were provided at the top and bottom of the column to ensure uniform load distribution and to avoid any stress concentration. The experimental observation recorded the nature of failure, axial deformation, and ultimate load. The experimental setup is shown in Figure 7.

## 6. Results

### 6.1. Compressive Strength Test of Unconfined Concrete

The compressive strength test results for unconfined concrete cylinders are summarized in Figure 8. It is evident from the test results that steel fibers have a pronounced effect on the strength and ductility of unconfined concrete. An increase is noted in both the strength and ductility of concrete samples. The percentage increase in strength is shown in Figure 8. The compressive strength of concrete increases from 17.64 MPa to 24.6 MPa with the addition of steel fibers from 0% to 2.5%, respectively. The highest value for strength was noted for mix G5 (2.5% steel fibers), which is 38.97% as compared to mix G1 (0% steel fibers). The results obtained in this study were correlated with the findings of previous literature [39], which deduce that the addition of steel fibers has increased the compressive strength of concrete. The increase in strength may be attributed to the bridging mechanism provided by the steel fibers. However, compressive strength may depend on other factors, such as the size, shape, aspect ratio, volume fraction, orientation and the surface characteristics of fibers, the ratio between fiber length and maximum aggregate size, the volume ratio between long and short fibers, and the concrete class [40]. Moreover, a 5.1% decrease in compressive strength was noted for the G6 (24.6 MPa) concrete mix as compared to G5 (16.8 MPa). The main reason for decreasing concrete compressive strength by increasing steel fiber may be the harshness of the concrete mix, as evident from the results reported in Table 4. It is further mentioned, that the addition of steel fibers leads to a greater surface area for the surrounding concrete which is not sufficient to develop proper bonding amongst the materials to facilitate the bridging in order to prevent micro-cracking. Literature [41] also suggested that the compressive strength of concrete decreased beyond the optimum dosage of steel fibers due to physical hindrance in obtaining a homogeneous concrete mix. It was found that by increasing the steel fibers contents, axial deformation increased from 1.8 mm to 5.36 mm (G1 to G4 concrete mixes) at the rupture point, considerably enhancing the ductility of concrete shown in Table 6, and the axial deformation is decreased from 5.36 mm to 5.04 mm (G4 to G6 concrete mixes). The stress–strain curve in Figure 9 and Table 6 shows that the G5 has the highest toughness as compared to other concrete mixes. The increase in ductility and toughness is due to the effective confinement contribution by steel fibers which prolongs the material failure. Similar findings were reported by literature [39] that steel fibers enhanced the fracture properties of concrete.

### 6.2. Axial Behavior of Column

Axial behavior parameters of short RC square columns are presented in Table 7 and Figure 10. The ultimate axial load-carrying capacity of the column is enhanced with an increase in the confinement of the column due to the addition of steel fibers and reducing the spacing of ties. The axial behavior of column specimens C2 and C3 are compared with the reference column specimen, C1. It was noted that the axial capacity of the C2 column was increased by 11% (443.4 kN) while that of C3 was increased by 55% (620.8 kN) as compared to C1 (399 kN). This indicates that the addition of steel fibers has a significant improvement in the axial behavior of RC column compared to reducing ties spacing, on the other hand. The increased axial capacity values are associated with the bridging mechanism and improved confinement effect of steel fibers. Thus, the inclusion of steel fibers was more effective in enhancing the load-carrying capacity of the column instead of reducing the column ties spacing. Similarly, C3 has a 67% increased toughness while C2 has a 35.8% increased toughness as compared to C1. This additional concludes that steel fibers have a more pronounced effect on energy absorption than reducing the ties spacing of RC columns. On the other hand, axial deformation of the C3 column increased by 7.4% and that of the C2 column increased by 22.2% as compared to C1. This indicates an adverse effect of the addition of steel fibers on axial deformation and ductility of RC columns. The findings of the current study are found to be in line with similar studies in the literature [42,43].

### 6.3. Failure Modes

The failure pattern of RC columns with or without steel fibers showed almost similar behavior. However, the column having steel fibers (C3) showed brittle behavior and sound cracking as compared to C1 and C2 columns. With the progression of the axial load on the column specimens, spalling of the concrete cover was observed in C1 and C2, while C3 column specimens exhibit prolong spalling behavior, as shown in Figure 11. At the ultimate axial load, crushing of concrete was observed in all three columns (C1, C2, and C3) specimens with complete exposure of longitudinal reinforcement to the atmosphere in the case of the C1 and C2 columns.

## 7. Conclusions

In the present research work, steel fibers were used for the enhancement of confinement and strength characteristics of reinforced concrete columns. Based on the experimental study, the following conclusions have been derived:In unconfined concrete, 0%, 1%, 1.5%, 2% and 2.5% steel fiber amounts were used in different mixes. An increase in the percentage of steel fibers increased the compressive strength; however, beyond 2.5%, the compression strength decreased. Thus, 2.5% of steel fibers is recommended as an optimum replacement proportion.The use of an optimum replacement ratio of 2.5% has resulted in an increase in the compressive strength by 38.97% as compared to unconfined concrete, accompanied by a substantial increase in ductility. It was further noted that beyond 2.5%, the addition of steel fibers adversely affects the strength and durability of concrete.In the case of confined concrete, the addition of steel fibers sufficiently increased the strength alongside a pronounced effect on the ductility of reinforced concrete. The confined increased in terms of an increase in concrete strength; however, a detailed study is needed to evaluate its effectiveness in reducing the spacing of stirrups.The current work investigated the columns on a reduced scale. Future work is needed on actual column sizes. Moreover, with recent advancements in the field of artificial intelligence [44,45,46,47], regression models shall be developed to select the most content of steel fibers in the column.

## Figures and Tables

**Figure 1 materials-15-03219-f001:**
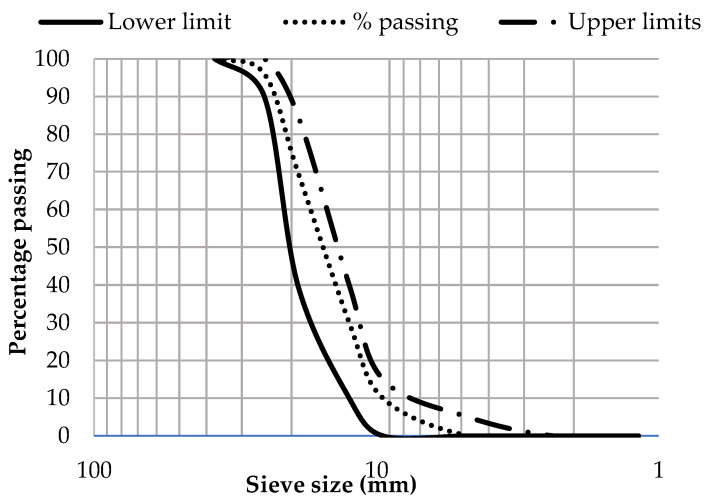
Gradation curve of coarse aggregate.

**Figure 2 materials-15-03219-f002:**
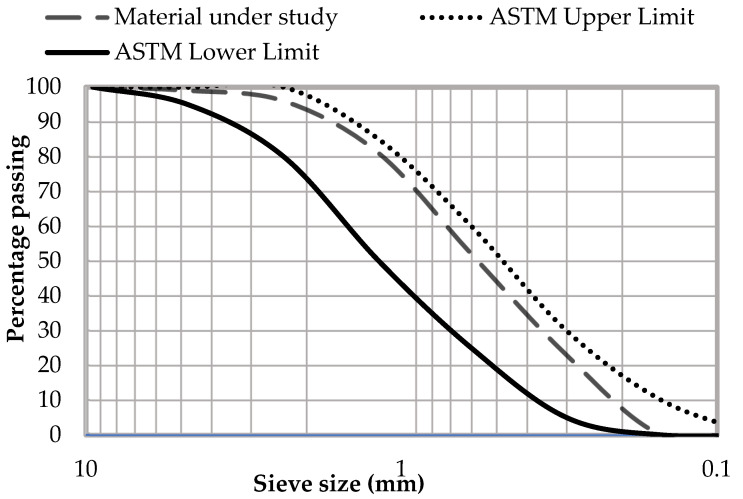
Gradation curve of coarse aggregate.

**Figure 3 materials-15-03219-f003:**
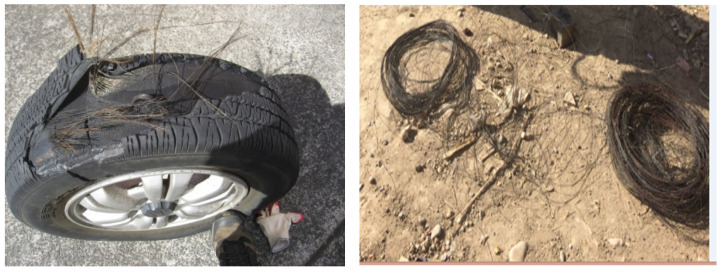
Steel fibers extracted from used tyres.

**Figure 4 materials-15-03219-f004:**
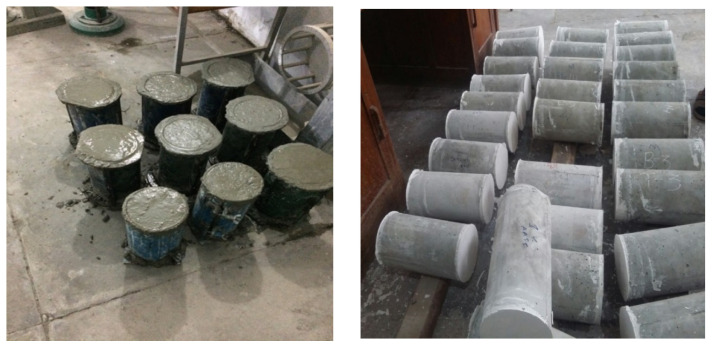
Cylindrical specimen of plain cement concrete and fibers reinforced concrete.

**Figure 5 materials-15-03219-f005:**
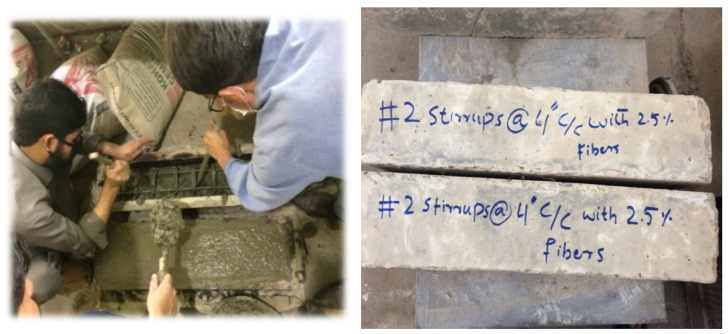
Casting of column specimen with and without fibers reinforcement.

**Figure 6 materials-15-03219-f006:**
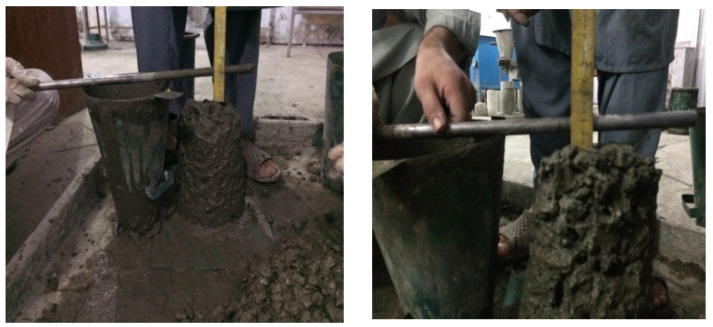
Slump cone test of fresh concrete.

**Figure 7 materials-15-03219-f007:**
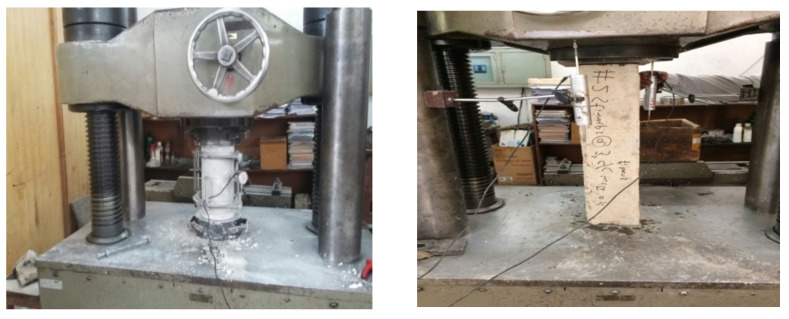
Cylindrical and column specimen testing in universal testing machine.

**Figure 8 materials-15-03219-f008:**
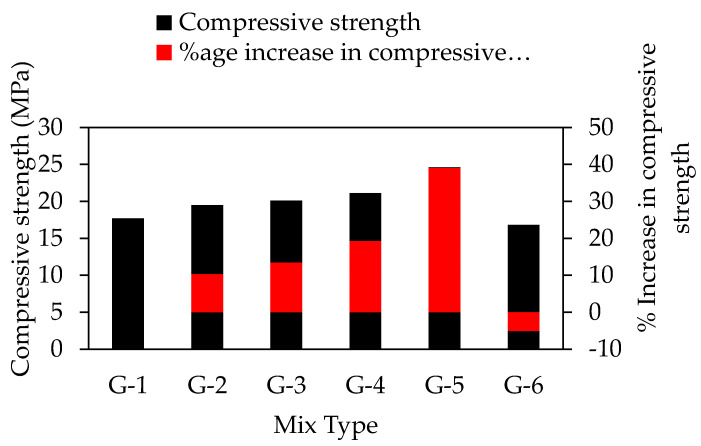
Compressive strength test result of cylindrical specimens and percentage increment in compressive strength.

**Figure 9 materials-15-03219-f009:**
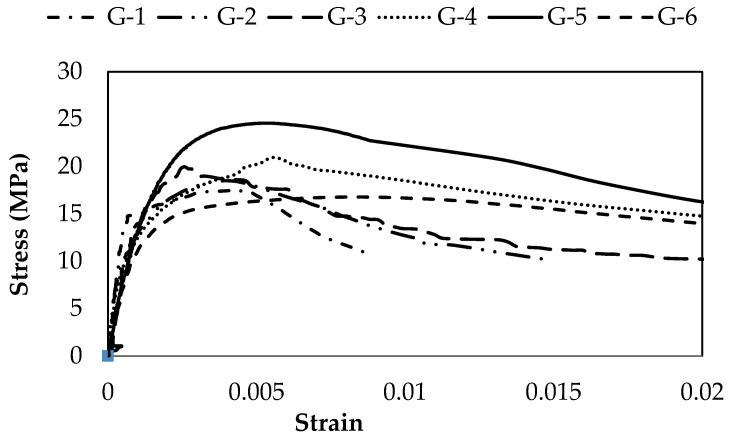
Stress–strain curves for unconfined concrete with steel fibers.

**Figure 10 materials-15-03219-f010:**
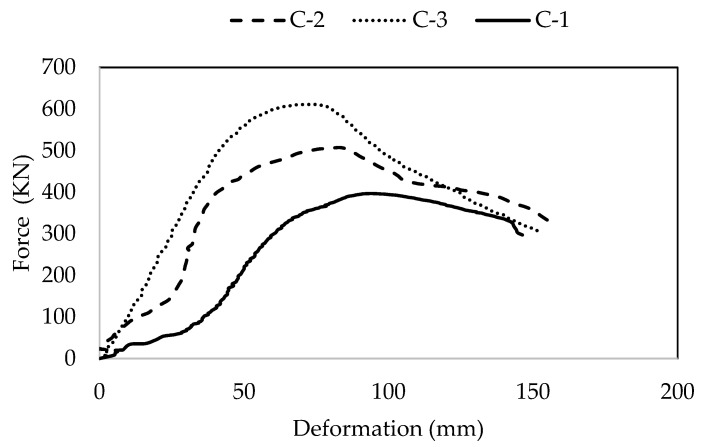
Force-deformation curves of confined concrete specimen.

**Figure 11 materials-15-03219-f011:**
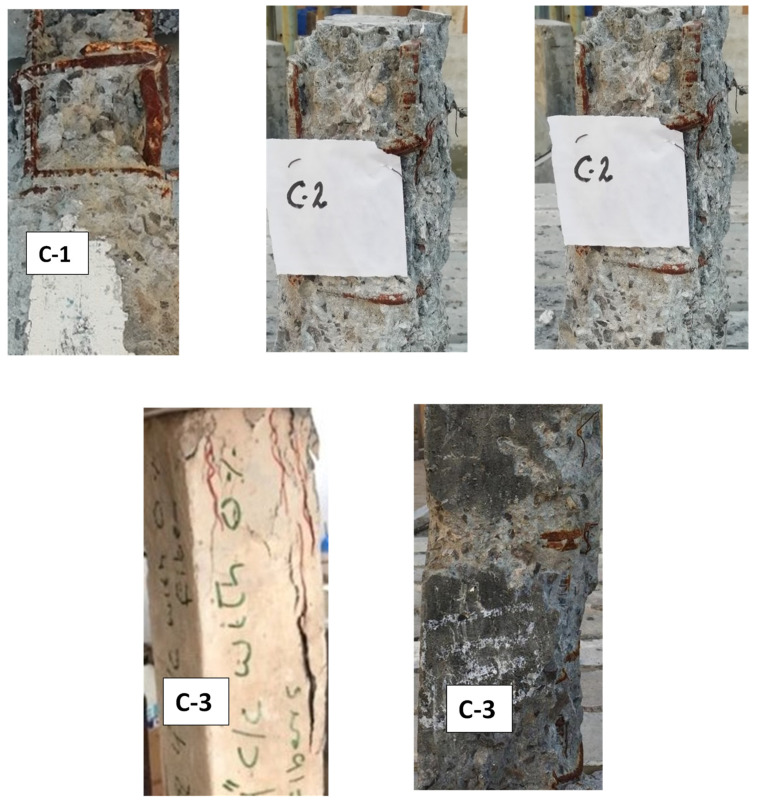
Failure modes of column specimens.

**Table 1 materials-15-03219-t001:** Properties of Portland cement.

Property	Specific Gravity	Fineness Modulus	Initial Setting Time	Final Setting Time
Value	3.15	345 m^2^/kg	45 min	250 min

**Table 2 materials-15-03219-t002:** Properties of aggregate.

	Specific Gravity	Bulk Density	Absorption Capacity	Max. Aggregate Size	Fineness Modulus
Coarse aggregate	2.63	1555.8 kg/m^3^	0.45%	25 mm down	------
Fine aggregate	2.30	------	1.50%	-------	2.40

**Table 3 materials-15-03219-t003:** Properties of steel fibers.

Property	Tensile Strength	Diameter	Length	Aspect Ratio
Value	132 kg/mm^2^	1 mm	75 mm	75

**Table 4 materials-15-03219-t004:** Mix design/proportion detail.

Water/Cement	Weight of Concrete Ingredient (lbs./yd^3^)				Proportions	Percentage of Steel Fibers	Number of Specimens
	Water (lbs.)	Cement (lbs.)	Fine Aggregates (lbs.)	Coarse Aggregates (lbs.)			
0.66	3.4	5.15	11.90	21.90	1:2.3:4.3	0	3
0.66	3.4	5.15	11.90	21.90	1:2.3:4.3	1	3
0.66	3.4	5.15	11.90	21.90	1:2.3:4.3	1.5	3
0.66	3.4	5.15	11.90	21.90	1:2.3:4.3	2	3
0.66	3.4	5.15	11.90	21.90	1:2.3:4.3	2.5	3
0.66	3.4	5.15	11.91	21.91	1:2.3:4.3	3	3

**Table 5 materials-15-03219-t005:** Slump cone test results.

Water/Cement	Slump Type	Slump Values (mm)	
		Control Samples	Percentage of Steel Fibers
0.65	True	89	0
0.65	True	76	1.5
0.65	True	64	2.0
0.65	True	50	2.5
0.65	True	41	3.0

**Table 6 materials-15-03219-t006:** Axial behavior of unconfined concrete.

Designation of Cylinder	Ultimate Load (KN)	Axial Deformation (mm)	Toughness/Energy Absorption KN-mm
G1	327.12	1.804	590
G2	354.24	3.174	1124
G3	394.44	4.86	1917
G4	408.5	5.36	2190
G5	438.56	5.29	2320
G6	320.26	5.04	1614

**Table 7 materials-15-03219-t007:** Axial behavior of confined concrete.

Column Type	Ultimate Load (KN)	Axial Deformation (mm)	Toughness/Energy Absorption (KN-mm)
C1	398.96	7.248	2892
C2	443.4	8.856	3927
C3	620.76	7.786	4833

## Data Availability

All the data used in the city is properly reported within the text.

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
