# Peer review of "Enhancement of Confinement in Scaled RC Columns using Steel Fibers Extracted from Scrap Tyres"

_materials, 2022, doi:10.3390/ma15093219_

Round 1
Reviewer 1 Report
This study investigates the effect of confinement in reinforced concrete column using the steel fibre which is derived from the scrap. The article requires some modification based on my comments as follows.
- Abstract: The findings in the abstract highlighted only the compressive strength of concrete. The key findings of column behaviour should be highlighted in the abstract.
- Line 119, 120, 126, 129, 134 add citation for the ASTM standards.
- Line 109-110, adding the steel fibre doubles the cost of construction, these fibres are less suspectable to the corrosion, available with designed strength which increase the strength of concrete. However, the steel fibre derived from scrap is suspectable to the corrosion. How this problem can be eradicated?. Please discuss this in the manuscript.
- Please maintain the consistence of the units (for example Fineness modulus in Table 1 in terms of kg/m2, bulk density in Table 2 in terms of b/ft3
- Table 3, how did the aspect ratio of steel fibre was calculated? Aspect ratio which defines the ratio of length and diameter of the fibre. Please check the value.
- Line 143-144, “The steel fibers were subjected to various tests for the determination of their mechanical properties” what are the test conducted to evaluate the properties of steel fibre?
- Line 155, provide proper citation for ACI standard.
- In general, the addition of steel fibre in the concrete is limited to 2% by volume due to the workability issues and conglobation. Why the steel fibres dosage has chosen upto 3% in Table 4? What are the problems faced while mixing the concrete due to the over dosage of fibres?
- Why the square column is adopted in this study? Is there any scientific reason behind this selection?
- Line 196-197, Include the Table number for “Slump cone test results”.
- Generally, Increasing the fibre content results in decreasing the slump value. This trend is reverse in Slump cone test results, justify this.
- Figure 9 should be improved and can be merged. Also, add axis title for both X and Y axis.
- Explain the reason for increasing the compressive strength by adding fibre upto 2.5% and decreasing beyond 2.5%. Correlate the results with the earlier studies.
- The discussion of compressive strength should be improved with more comparison.
- The discussion section is very weak.
- Conclusions are technically sound.
- English language should be checked thought the manuscript.
Author Response
Attached in word format.

Reviewer 2 Report
The paper ”Enhancement of Confinement in RC Columns Using Steel Fibers Extracted from Scrap Tyres” is suitable to be published in Materials Journal just after some major updates.
Regarding the fact that the authors want to publish this research in Materials, they must add some SEM images and of the morphology of the concrete and XRD analysis for the determination of the phases.
Also, I want to see the morphology of the fibers.
Please revise figure 9.
Author Response
Attached is a word document.

Reviewer 3 Report
This article is interesting but needs some modifications to be suitable for this journal. The main comments are:
- The novelty of this paper should be further justified and to establish the contributions to the new body of knowledge.
- Abstract section should be improved considering the following structure: introduction, problem statement, methodology, results, and conclusion.
- In Introduction section, the authors should improve the research background, the review of significant works in the specific study area, the knowledge gap, the problem statement, and the novelty of the research.
- Introduction section could be better organized. I cannot readily see the significance of this study compared to the previous works. The literature review should be extended to more recently published works available in the literature. Some of the papers which are directly related to this study are:
(a) Influence of super absorbent polymer on mechanical, rheological, durability, and microstructural properties of self‐compacting concrete using non‐biodegradable granite pulver, Structural Concrete 22, E1093-E1116
(b) Effect of super absorbent polymer on microstructural and mechanical properties of concrete blends using granite pulver, Structural Concrete 22, E898-E915.
- The presentation of the results and conclusions could be improved.
The article investigates the effect of confinement in reinforced concrete columns using the steel fibre which is obtained from the scrap.
The topic is interesting but the paper has a lot of flaws.
Not that novel unless the authors can highlight the novelty as per my detailed comments to the authors. No clear and evident conclusions were drawn from the paper.
The content needs more information. The tables and figures are ok.
there are any comments regard content that can be improved.
Author Response
Attached is a word document.

Reviewer 4 Report
The paper presents an Enhancement of Confinement in RC Columns Using Steel Fibers Extracted from Scrap Tyres. The theme of the contribution is not new. There is a lot of effort in the engineering and research community to enhance concrete and concrete structure bearing capacity. Many experiments and publications were done and presented in this area. The main benefit of the paper is the use of steel fibers extracted from scrap tyres in the confinement of reinforced concrete. It would be beneficial not only in building engineering but also in environmental engineering and ecology. So, it is worth paying attention to.
The paper written in high-level English is well structured. Graphs, tables, and figures presented are clear, and they illustrate the problem solved
appropriately. The theme is contemporary. Several mistakes and typos were found in the text. I recommend the paper for publication after a minor revision.
Comments:
- page No. 2, lines 78 and 82: "External" and "Rehabilitation" should be written with lower case letters
- page No. 2, line 88: missing comma in "300,000 000"
- page No. 3, line 113: upper case letter in "Hence, Steel fibers"
- page No. 3, line 117: The subsection title is written in bold. Should be italic
- page No. 5, Table 3: The length of fiber is 3 mm. It can be convenient to add a note that the fibers were cut.
- page No. 5, line 152: "traverse reinforcement" -> "transverse reinforcement"
- page No. 6, lines 177 to 187: The paragraph is italic and connected with the title.
- Figures 8 and 10: "Constitutive Material Model" should be changed to "Stress-strain diagram" or "curves".
Author Response
Attached is a word document.

Reviewer 5 Report
The paper deals with an experimental investigation regarding the effect of the addition of tyres-extracted fibers on the compression behavior of concrete, also in presence of reinforcement.Very few results can be found regarding the behavior of SFRC reinforced columns under compression, even at reduced scale. Hence, the results are worth to be published. However, the paper should be significantly revised according to the comments below in view of providing sufficient details and insight.
1. Line 67: the statement “It has been demonstrated that FRC increases the confinement of concrete in columns and also maintains the continuity between core and cover concrete.” is not a general one and it should be supported by providing proper references.
2. Line 84. How is “corrosion of steel plates” related to the application of externally bonded FRP?
3. Line 85: Please, provide specific references and describe more clearly under which conditions confinement can be beneficial. Campione et al ( [27]- check also the name of the reported journal) tested small specimens, not columns. The work by Bharti et al. [26] is focused on the beam-column joints, where the dominant action is shear; their results can be hardly extended to the behavior of columns under compression. The reference for Sidney Mindess et al. is not reported. None of the reported references refer to real-size columns, hence the statement “In internal confinement synthetic fibers such as steel fibers, polypropylene fiber, glass fibers have been used in concrete to enhance RC column confinement and to control microcrack propagation by the bridge mechanism.” is unsupported. The entire section should then be reviewed.
4. Following the previous comments, the presented work does not focus on the behavior of real size columns, but of scaled specimens. The two things are obviously completely different, being the behavior of real-size columns governed by a number of factors as strain distribution along the section, scatter of material properties, effects of eccentricity of the applied load et alia. Consequently, the title should be reviewed as “Enhancement of Confinement in Scaled RC Columns Using Steel Fibers Extracted from
Scrap Tyres” and the limitations of the investigation due to the specimens’ size should be properly reported and discussed.
5. #2.4. the use of fibers extracted from tyres is among the interesting aspect of the reported investigation. Consequently, authors should describe the process of extraction, which type of controls were carried out on the dimensions of the fibers, if any selection process were carried out and, finally, which mechanical tests were carried out and their results. In absence of such information, reproducibility of results would be not guaranteed.
6. #4.1 and 4.2. The considered amount of fibers is very high (up to 3.0% Vf). Under such circumstances, mixing and casting operation could be very difficult. Authors should report any issue to this regard and specify if specific measures were taken to assure proper flowability. Additionally, the casting operation may induce fibers orientation. Did authors consider specific measures to such regard? Did authors checked fibers orientation in the final specimens? Fiber orientation (and/or fiber grouping) could provide some background for the results of compression tests.
7. #4.2. Details regarding bar sizes, position and mechanical properties are missing and they should be reported.
8. #5.2. Several details regarding the test setup are missing, as: length of basis of measurement for LVDTs, type and class of LVDTs and load cell, end constraints (fixed/rotating plates), load increment with time. Additionally, Figure 07 shows some LVDTs were used to measure top loading plate displacement. If such measurement is used to derive the ‘strain’ adopted in Figure 10, then such definition of strain is an unproper one. In fact, the adopted measurement accounts for several effects, as stress diffusion at the
loading ends, machine elongation (and others) and it cannot be used to derive a local property as strain. Consequently, it would be better to refer, in both Figure 10 and discussion, to “total specimen shortening” instead of “strain”.
9. 6#.2. The entire section should be redraft. All the results should be reported in a table (and not only a few curves related to three selected specimens). Additionally, some description of the specimens at failure (and maybe some pictures as well) should be incorporated to discuss if and how the presence of fibers led to different failure mechanism, potentially associated to changes in the crack patterns.
10. General editorial remark: please, be sure that all the cited standards are properly reported in the references and do not mix Metric and Imperial units. Please, also do not use kg or tons instead of N (or kN). Finally, conclusions should be completely revised accounting for all the previous comments
Author Response
Attached is a word document.

Round 2
Reviewer 1 Report
The article has improved to a greater extent.
Author Response
Attached is a word document.

Reviewer 2 Report
The paper ”Enhancement of Confinement in RC Columns Using Steel Fibers Extracted from Scrap Tyres” can now be published in present form.
Author Response
Attached is a word document.

Reviewer 3 Report
Thanks for addressing my comments. The paper can now be accepted for publication.
Author Response
Attached is a word document.

Reviewer 5 Report
In general, reviewer's comments were addressed. However:
- comment #8. The required information regarding the test setup should be reported. Additionally, defining 'strain' on the basis of gauge length (as confirmed by the authors) is a conceptual mistake and Figures should refer to 'axial shortening' only.
- comment #9. Authors should make some additional effort to incorporate description and pictures of the specimens at failure.
Author Response
Attached is a word document.
